

# Gross motor coordination and their relationship with body mass and physical activity level during growth in Children aged 8–11 years old: a longitudinal and allometric approach

Matteo Giuriato[1,2,3], Nicola Lovecchio[4], Vittoria Carnevale Pellino[5,6], Jan Mieszkowski[3], Adam Kawczyński[7], Alan Nevill[8] and Valentina Biino[1,2]

[1] Department of Neurosciences, Biomedicine and Movement Sciences, University of Verona, Verona, Italy
[2] Department of Human Science, University of Verona, Verona, Italy
[3] Gdansk University of Physical Education and Sport, Gdańsk, Poland
[4] Department of Human and Social Sciences, University of Bergamo, Bergamo, Italy
[5] Laboratory of Adapted Motor Activity (LAMA), Department of Public Health, Experimental Medicine and Forensic Science, University of Pavia, Pavia, Italy
[6] Industrial Engineering, University of Tor Vergata, Rome, Italy
[7] Wrocław University of Health and Sport Sciences, Department of Paralympics Sports, Wrocław, Poland
[8] Faculty of Education, Health and Wellbeing, Walsall Campus, Walsall, UK, University of Wolverhampton, Walsall, United Kingdom

Corresponding author
Vittoria Carnevale Pellino,
vittoria.carnevalepellino@unipv.it

## ABSTRACT

**Background:** Gross motor coordination (GMC) is fundamental to the harmonious development of physical skills during the growth of children. This research aimed to explore the developmental changes in GMC during childhood, having controlled for the differences in children's body size and shape using a longitudinal, allometric scaling methodology.

**Methods:** A total of 104 children from North-East Italy of third-fourth- and fifth-grade students and each participant were tested for three consequently years. Subjects performed the short version of korperkoordinationstest fur kinder (KTK3) and the anthropometric characteristics were simultaneously collected. The physical activity questionnaire (PAQ-C) was used to evaluate the weekly physical engagement.

**Results:** Ontogenetic multiplicative models suggested nonlinear associations with GMC across the children's developmental years. Linear physique was a significant predictor associated with the development of GMC across time. PAQ-C was significantly associated with GMC from 8 to 13 years old.

**Conclusion:** Growth does not respect linear trends. A greater lean body mass could be assessed as a reliable predictor of GMC in children. The crucial period of life to improve the GMC is 8–10 years, while stabilization becomes evident from 10 to 13 years.

# INTRODUCTION

Physical activity (PA) practice becomes crucial for the general development of children during growth phases whereas the ability to perform different motor action is defined motor competence (MC; *Henderson & Sugden, 1992*). In particular, the MC within childhood is often defined as gross motor coordination (GMC; *Robinson et al., 2015*) because reflects the ability to perform a range of fundamental movement skills (*e.g.*, running, jumping, hopping) that are fundamental as achievements to satisfy the demands of home, sport, school, and the social environment interaction. The perception of MC could contribute to start and continue different training programs or sport activities in children maintaining adherence through years (*De Meester et al., 2016*). In turn, the development of GMC lead to more specialized movement sequences, such as sport-specific gestures (*Clarke & Metcalfe, 2002*) (*e.g.*, throwing in basketball) and lifelong movement skills (*e.g.*, cycling and swimming) (*Hulteen et al., 2015*). In fact, GMC, which requires large muscles for balance, limbs, and trunk movements, become essential to gain more complex movements also, useful after childhood (*van der Fels et al., 2019*).
The achievement of GMC is not an automatic process but needs a planned education and practice (*MacNamara, Collins & Giblin, 2015*).

*Barnett et al. (2016)* suggesting that variables such as age, sex and weight status affect GMC while *Queiroz et al. (2020)* found a positive modulation and improvements of GCM (in addition to body mass index (BMI)) in young belonging to a family with a high socio-economic-status family that permit higher possibilities to access in environmental contexts, sports facilities or sports clubs. These results underline the connection between motor practice and central nervous system development (*Davids, Button & Bennett, 2008*) where the evaluation of GMC is often used as an indirect measure of healthy development (*Barnett et al., 2016*). In fact, longitudinal research provides a profound overview of adaptations and evolution during tumultuous phase for body mass, weight, and gender hormonal modifications both for girls and boys (*Henrique et al., 2018*; *Lima et al., 2017*), whereas longitudinal studies with at least three measures separated by time provide a more rigorous interpretation of motor development (*Baxter-Jones, 2017*). It appears evident that early physical experiences and those timing and structured time are necessary to guarantee the development of basic movements that allow children to obtain practical skills (*MacNamara, Collins & Giblin, 2015*). Considering children during the growth phases, *Dos Santos et al. (2018)* and *Armstrong & Welsman (2019)* suggested an allometric multilevel modeling analysis between the years (longitudinal) starting a new analytical approach to developmental exercise assessment (*Aitkin, Anderson & Hinde, 1981*). Allometric permits to analyse height and body mass as two different covariates.

Further, *Dos Santos et al. (2018)* showing that GMC is influenced from Body mass ($p < 0.001$) and stature ($p < 0.001$) negatively and positively respectively. Suggesting that ectomorphic (more linear children) in their overall body shape, and less heavy, show the best GMC development across time.

Indeed, it permits to understand the influence of variables (*i.e.* age, body mass, etc.) within an allometric framework to feed a flexible and sensitive evaluation of exercise performance variables (*Nevill et al., 2009*). For example, *Lovecchio et al. (2019)* and *Giuriato et al. (2020)*, through allometric modeling, show that the optimal body shape associated with physical performance (lower limb explosive strength, trunk strength and endurance performance) is a linear physique: slim with a high level of lean body mass. Further, *Dos Santos et al. (2018)* suggested that leaner and physically fitter children tended to show better results of GMC. In point of this, *de Niet et al. (2021)* suggested that physical education (PE) teachers, trainer, coaches or physical educator need to be conscious of the level of affection of the growth on GMC. Moreover, previous studies identified the importance to evaluate the PA level as a precursor of GMC development of children because it could better predict the motivation to start different activities, the motivation to acquire different motor skills and maintaining adherence to the PA programs (*Khodaverdi et al., 2016*; *Hulteen et al., 2020*).

In children, the GMC were often assessed trough the Körperkoordinationstest Für Kinder (KTK) that is a reliable and low-cost battery field tests (*Kiphard & Schilling, 2007*) consisting of four standardized measurements such as walking backwards, jumping sideways, moving sideways, and hopping for height. Recently, *Novak et al. (2017)* proposed a short version of standard KTK: the KTK3 that showed a strong correlation with the KTK scores (r = 0.98, $p < 0.001$). In particular, the KTK3 seems to be more applicable in sports and school settings because the hopping for height test is more time-consuming and with higher risk of injuries (*e.g.*, ankle sprain) than other sub-tests (*Novak et al., 2017*; *Platvoet et al., 2020*). Indeed, KTK3 administration time is about 10 min against 20 min of KTK (*Novak et al., 2017*).

To the best of our knowledge, no previous studies investigated the relationship between KTK3 results and height and body mass using an allometric modelling that permits an accurate evaluation of children body shape and performance in accordance with their growth. We hypothesized that a better comprehension of the relationship between these variables could help PE teachers and sports trainers to better evaluate their children's capacities and propose the adequate activities to their real growth.

Thus, adopting allometric modeling to investigate in better manner the influence of body mass and height separately on GMC; the aim of this study was focused on a longitudinal investigation (three consecutive years) of the GMC levels within the relationships between height and body mass using the allometric scaling analyses and the relationship of PA level with the GMC development in samples of children aged from 8 to 11 years old.

## MATERIALS AND METHODS

### Participants and study design

Considering children during the growth phases, *Dos Santos et al. (2018)* and *Armstrong & Welsman (2019)* suggested an allometric multilevel modeling analysis between the years (longitudinal) starting a new analytical approach to developmental exercise assessment (*Aitkin, Anderson & Hinde, 1981*). This longitudinal study involved 104 children aged

8–11 years that were recruited from primary and middle schools in the north-west region of Italy from their PE teachers in the 2017. Every group were assessed for three consecutive years from 2017 to 2019. All groups that started evaluations at the beginning of October 2017 attended third, fourth, fifth, and sixth grade. The authors chose the month of October because the school had just begun, and children had not yet participated in PE program that could alter their performances. Children with known chronic cardiac, respiratory, neurological, or musculoskeletal disorders were excluded from the study while all those were physical active during PE classes and with a contextual medical certificate were included to the assessment. They have taken all the measures during the PE lessons as scheduled in the morning framework (8.00–12.00 AM). The study protocol, including each feature of the experimental design, was approved by the Institutional ethical boards of schools (Prot. 1523; Cod 123.1/6) in accord with the "World Medical Association Declaration of Helsinki" as revised in 2018. According to the consent, all participants were free to withdraw their participation at any time, *a priori*, signed by parents or legal guardians.

## Procedure

Data collection, including demographic/anthropometric information (sex, age, body mass, height), were collected in a separate appointment before the test sessions using standardized techniques from the sport scientist specialists in accordance with PE teachers. The age of the children from the birth date was calculated and subsequently rounded down values (*World Health Organization, 2008*).

## Anthropometrics

Height was measured through a portable stadiometer with a precision of ± 1 mm (Stadiometer Seca 213, Intermed S.r.l. Milan, Italy) with children in an upright position, barefoot placed slightly apart, arms extended, and head positioned in Frankfort plane. Body mass was assessed through a beam scale with a precision of ± 100 g (Seca 813: Intermed S.r.l., Milan, Italy), with children wearing light clothing, without shoes and stood upright at the center of the platform of the mass scale. Then, the Body Mass Index (BMI) were calculated using the formula: weight (kilograms)/height $^*$ height (meters).

## Gross motor coordination

GMC was evaluated using a new and short version of the KTK battery (*Kiphard & Schilling, 2007*) with a test-retest reliability coefficient of 0.97 (*Kiphard & Schilling, 2007*). Further, KTK3 retains relatively high accuracy and presents the potential to minimize the risk of injury, such as ankle sprain, to children by removing the hopping for height component (r = 0.98, $p < 0.001$). So, the other three subtests combined involve, in total, 10 min increasing the practicality of the battery and the full compliance of children.

In particular, the KTK3 procedure applied consisting of three trials:

1. Walking backwards on balance beams (three different widths, 6, 4.5 and 3 cm): A maximum of 24 steps (eight per trial) were counted for each balance beam, which comprises a maximum of 72 steps (24 steps × 3 beams) for this test.

2. Moving sideways as quickly as possible using two boxes (25 cm × 25 cm × 5.7 cm) per two times, transferring the first plate, stepping on it, etc. The number of relocations was counted and summed over two trials.

3. Jumping sideways over a board (60 cm × 4 cm × 2 cm) as many times as possible in 15 s per two times. The number of jumps over two trials was summed.

Trained supervisors carried out all assessments in the same gym school context. All participants receive a demonstration and verbal instruction for the KTK3 battery. The presence and collaboration of the curricular PE teachers were guaranteed to meet the students' confidence.

## Physical activity questionnaire

PA was assessed using the Physical Activity Questionnaire for older children (PAQ-C) questionnaire Italian version (*Gobbi et al., 2016*). The PAQ, as the version of IPAQ for older children, have acceptable reliability (males, r = 0.75; females, r = 0.82) and convergent validity (*Crocker et al., 1997*) and the administration and scoring are based on a self-administered, 7-day recall questionnaire (nine items). The nine items were proposed as follow:

1. Physical activity in your spare time: Have you done any of the following activities in the past 7 days (last week; *e.g.* skipping, rowing, running, bicycling etc.)? If yes, how many times (from "no" to "seven times or more")?

2. In the last 7 days, during your physical education (PE) classes, how often were you very active (playing hard, running, jumping, throwing)? From "I do not PE" to "always"

3. In the last 7 days, what did you do most of the time *at recess*? From "sat down" to "run and played hard most of the time"

4. In the last 7 days, what did you normally do *at lunch* (besides eating lunch)? From "sat down" to "run and played hard most of the time"

5. In the last 7 days, on how many days *right after school*, did you do sports, dance, or play games in which you were very active? From "none" to "five times"

6. In the last 7 days, on how many *evenings* did you do sports, dance, or play games in which you were very active? From "none" to "six or seven times"

7. *On the last weekend*, how many times did you do sports, dance, or play games in which you were very active? From "none" to "six or more times"

8. Which *one* of the following describes you best for the last 7 days? From "all or most of my free time was spent doing things that involve little physical effort" to "high very often did physical things in my free time"

9. Mark how often you did physical activity (like playing sports, games, doing dance, or any other physical activity) for each day last week. From "none" to "very often"

Collected data on participation in different activities and sports, effort during PE classes, and exercise during the past 7 days (lunch, outside school, evening, and weekend).

The average overall score (from one to five for all of the nine items, with five representing a very physically active lifestyle) was calculated to determine each subject's PAQ-C score.

## Data and statistical analysis

Data were manually collected by two operators and then inserted in separated sheets supporting anthropometric data with raw data of KTK3. All students are managed with a personal code to guarantee anonymity and a quick resume to match the two subsequent measurements. An appropriate method of analyzing longitudinal (repeated measures) data is to adopt a multilevel modeling approach, an extension of ordinary multiple regression where the data have a hierarchical or clustered structure (measurements grouped at different levels). For example, the repeated measure data, where individuals are measured more than one occasion represents hierarchical or clustered structure. In this study, children assumed to be a random sample represent the level-2 units, with the children's repeated measurements recorded at three visit occasions, being the level-1 unit. After this longitudinal allometric approach, we assessed the influence of body mass on GMC.

*Nevill et al. (1998)* suggested using the ontogenetic multiplicative model (*Nevill et al., 1998*) where $y = \text{body mass}^{k1} \cdot \text{stature}^{k2} \exp(a_i + b_i \cdot PA) \cdot e_{ij}$. A modified approach that considers body mass, height, and PA was used to compare our purposes and assumptions. Our model consists in a log-linear multilevel regression expressed as reported below:

$$\text{Ln}(KTK_{ij}) = k_1 \cdot \log_e(\text{Body Mass}) + k_2 \cdot \log_e(\text{Height}) + a_i + b_i \cdot PA + \text{loge}(e_{ij}) \tag{1}$$

In the previous formula, $k_1$ and $k_2$ are the ontogenetic allometric coefficients; $a_i$, and $b_i$ can vary randomly from different children (level-2), and $\log_e(e_{ij})$ is assumed to have a constant error variance between visit occasions (level-1). The constant $a_i$ can vary for different populations, such as the fixed factor gender, academic grade, and test through years.

All parameters of M1 were simultaneously estimated using complete maximum likelihood procedures implemented in the MLwiN (*Rasbash & Woodhouse, 1995*). These procedures are robust, efficient, and consistent optimizing the maximum likelihood if multicollinearity were present (*Hedeker & Gibbons, 2006*).

Additionally, the inverted body mass index (iBMI; 1,000/BMI; cm$^2$/kg) was calculated (Heigh$^2$/weight) as a valid marker of subject's lean body mass and then a better predictor of percentage body fat (*Nevill & Holder, 1995*; *Nevill et al., 2011*). Since iBMI has a positive linear association with percentage lean body mass it can represent an index of it.

## RESULTS

Multilevel modeling results are in Table 1. The model suggested the absence of difference between sex ($-0.001 \pm 0.018$, $p = 0.522$). GMC has a nonlinear trend across the study years (see Fig. 1). Further, this model shows the ontogenetic scaling factors that best define children's body size/shape and their GMC evolution from 8 to 13 years of age. Body mass ($-0.244 \pm 0.045$, $p < 0.001$) and stature ($0.644 \pm 0.194$, $p = 0.0005$) parameter estimates are statistically significant, suggesting that inverted body mass index (IBMI), a measure of lean body mass, in their overall physique, and less heavy, demonstrate the best GMC development through years. PA (using PAQ-C) was significantly associated with GMC

**Table 1 Multilevel results for gross motor coordination model based on class 3 (8-year) and male subjects.**

| Parameters | Level 1 | | | |
|---|---|---|---|---|
| | **Estimate B** | **± SE** | **Z-score** | **p-value** |
| Intercept | 0.009 | 0.002 | 4.50 | <0.001 |
| LnHeight | 0.644 | 0.194 | 3.32 | 0.0005 |
| LnBody Mass | −0.244 | 0.045 | −5.42 | <0.001 |
| Physical Activity | 0.52 | 0.012 | 43.33 | <0.001 |
| Sex (Female) | −0.001 | 0.018 | −0.06 | 0.5222 |
| Class 4 | 0.089 | 0.037 | 2.41 | 0.0081 |
| Class 5 | 0.178 | 0.042 | 4.24 | <0.001 |
| Class 6 | 0.215 | 0.058 | 3.71 | 0.0001 |
| Class 7 | 0.241 | 0.078 | 3.09 | 0.0010 |
| Class 8 | 0.261 | 0.089 | 2.93 | 0.0017 |
| Year of test 2018 | 0.083 | 0.027 | 3.07 | 0.0011 |
| Year of test 2020 | 0.142 | 0.053 | 2.68 | 0.0037 |
| Random variance components (random intercepts model) | | | | |
| Level 1 (within individuals) | | | | |
| Variance (eij) | | 0.009 ± 0.001 | | |
| Level 2 (between individuals) | | | | |
| Variance (uij) | | 0.009 ± 0.002 | | |

**Note:**
Male Subjects and Class 3 do not appears because they would be redundant Class 4 = 9-year; Class 5 = 10-year; Class 6 = 11-year; Class 7 = 12-year; Class 8 = 13-year.

changes from 8 to 13 years. The variance components show significant intra-individual differences in GMC changes through the years, different individual growth rates. Furthermore, GMC differs significantly from 8 to 13 years (Table 1).

The positive interaction between classes is significant, proposing that GMC development (trajectory over age) increases between the years (see Fig. 1).

## DISCUSSION

The main goal of the study is to evaluate the trend of GMC during growth spurt according to the anthropometric characteristics (height and body mass) and PA level on three consecutive years of school. The research was based on subjects aged 8 and 11 years old that followed a longitudinal observational approach (aged 10 to 13 years old) and, to make the study more reliable, it was used a multilevel allometric modeling to reach our aim (*Dos Santos et al., 2018*). Our results showed that the GMC was positively influenced by higher lean body mass and showed an increase performance across years without any difference between sex. The authors considered the importance of knowing the GMC trend through allometric approach to avoid the misleading conclusion where the improvements are related to the children' growth (*Giuriato et al., 2020*; *Sammoud et al., 2018*). Moreover, the allometric scaling becomes the most robust approach (Eq. 1) to compare GMC with anthropometric characteristics within the growth path variation

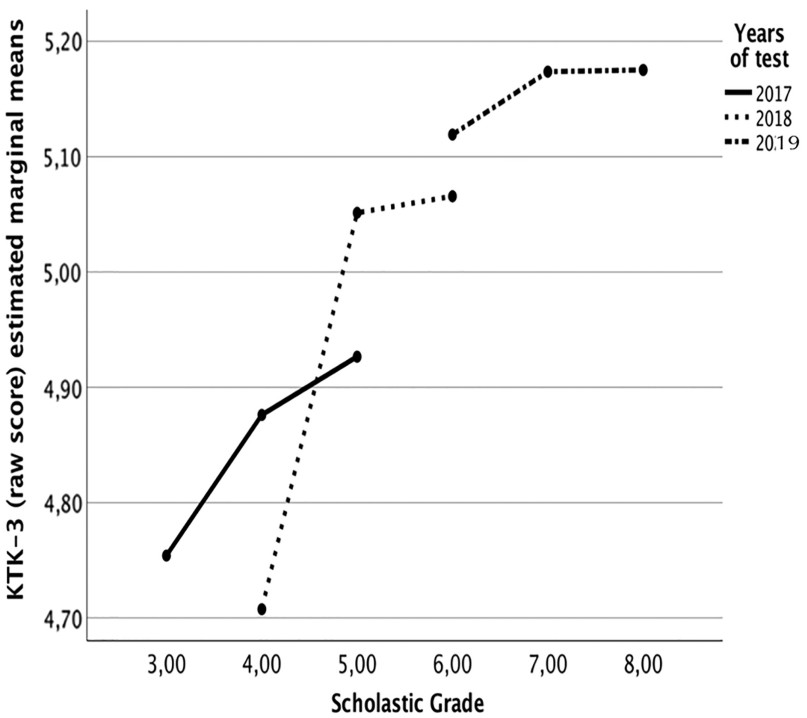

**Figure 1 Estimated marginal means of KTK-3 test per age (at 2008) during the three longitudinal step assessments.** Covariates appearing in the model are evaluated at the following LnHeight = 0.3890, LnMass = 3.7431.

(*Tsiotra et al., 2009*). Indeed, the GMC increase in third, fourth and fifth academic grades, ergo from 8 to 10 years old respectively (*i.e.*, the age at which pre-adolescence begins) (*Malina, Bouchard & Bar-Or, 2004*), while remain constant in other grades that correspond to the beginning of adolescence (11, 12, 13 years old) (Fig. 1).

Based on multilevel modeling analyses, we found that children with linear physique perform better in GMC during growth. Indeed, the allometric approach explains that ectomorphic body shape, *i.e.*, slim, with low body fat, is the best body shape to develop GMC during growing, based on ratio $Height^{0.644}$: Body $mass^{-0.244}$ (Estimated B column–Table 1). Our results demonstrated that lean body mass affects GMC across time. A proportional allometric model was applied to identify the influence of height and body mass. Multilevel modeling output across the years suggested a ratio, between the estimated B parameter, of 1:3 between height (0.644) and body mass (−0.244), respectively. This scaling method identified a tendency iBMI ratio between height and body mass (Table 1) with GMC: note that *Nevill & Holder (1995)* suggested that iBMI correspond to Lean Body Mass Index (LBM) because it has a positive linear association with percentage lean body mass (LBM% = 100 − BF%). Only one study investigating longitudinal GMC through allometric modelling (*Dos Santos et al., 2018*) and it did not wholly embrace our features. The authors identified the ectomorphic children as the optimal body shape for GMC outcomes suggesting that excess fat body mass negatively affected the performance. It may be possible that lean body mass promotes stabilization and propulsion of the body. Moreover, *Faigenbaum (2000)*. showed that regular

participation in strength-training programs during childhood is an excellent predictor of motor skills improvement. Also, *de Souza et al. (2014)*, with a retrospective study, demonstrated that the children who were fitter and more active at 10 years old correspond to those with higher GMC at 6 years old. We could suppose that lean body mass may be a precursor of adequate GMC performance and motor skills. On the other side, *Dos Santos et al. (2018)* revealed a significant difference between gender, as opposed to the results of our study. These discrepancies could be explained by the different databases used for the allometric scaling: our data was based on a raw score of KTK3. Indeed, *Dos Santos et al. (2018)* have an already standardized dataset (*Kiphard & Schilling, 2007*) (MQ values) where the natural trend is described through the longitudinal data of the class between the years. Furthermore, the lack of differences between males and females underlies an interesting finding from a practical perspective (Estimated B parameter: −0.001; $p = 0.522$). When the coordination has been assessed, the differences between sex are alleviated. Children generally decline their GMC with age (*Giuriato et al., 2019*). *Dos Santos et al. (2018)* showed a nonlinear trend in GMC across time through allometric modelling, suggesting a performance peak at nine years old. In our features, the exponent of age square remained negative in all models. To avoid invasive analysis, the field test like KTK3, becomes a valuable way to verify the correct development of GMC. KTK3 could be considered a "physical tool" to indirectly evaluate the brain and motor system developments (*Latash, 2012*).

Our study found also that the PA level influenced the GMC (Table 1). The estimated parameter B (0.52) suggested a positive influence of the PA level (investigated using PAC-C questionnaire) on the GMC through the years ($p = 0.001$). This result is in line with *Giuriato et al. (2021)*, which highlighted that extra school hours of sports practice (*i.e.* PA) were the first predictor of performance. Considering an heterogenous sample of subjects in their growth fase coming from school context, from a first view, the results about PA–GMC may seem obvious, because there is extensive documentation confirming that physical activity affects positively motor coordination (*de Souza et al., 2014*; *Fransen et al., 2014*). Indeed, adequate development of GMC through an satisfactory quantity of PA in youth may be important for the improvement and maintenance of health-related fitness from childhood to/into adolescence (*Stodden et al., 2014*). From another perspective, the Italian sample taking part in the study may not perform an adequate amount of physical activity GMC-based. In the light of this, allow opportunities to be physically active and build GMC during PE offering a valid possibility to children and adolescents to train in an appropriate manner (*Lai et al., 2014*; *Gao, Chen & Stodden, 2015*).

The KTK3 was composed of three items; two are based on jumping.

The strength of this longitudinal study is the allometric approach linked with multilevel modeling analyses and the use of KTK3 battery (*Vandorpe et al., 2011*). This study suggested some practical applications during the early stage of life. Firstly, during childhood, the trainer/PE teacher should be aware of the absence of differences between sex from 8 to 13 years old in GMC. Secondly, coaches should consider strong rationale during evaluation: sample-specific scaling model based on anthropometric characteristics

rather than theoretical models based on the assumption of body dimension similarity. Finally, the results suggest that this age group (third, fourth and fifth academic grade) has great potential to improve GMC in both sexes. Clearly, we cannot say whether this is the best niche (*Gallahue & Ozmun, 2006*). Considering this, an adequate PA level is essential to increase GMC, especially during the 8–10 years old period (third–fifth grade of school) as highlighted in Fig. 1 where the motor experiences modulate the patterns governing human movement (*Brown-Lum et al., 2020*; *Wrisberg et al., 1995*). This method is helpful for the first analyses. Still, the growth does not respect linear trends, and then a robust approach is necessary.

We are conscious that the study had some limitations such as the low numerosity of the enrolled sample and the evaluation of PA level through a questionnaire and not directly. Moreover, the body mass was calculated through IBMI and not considering perimeters and folds. But the substantial part of the study is the longitudinal approach through 3 years investigation and the multilevel modeling analyses that permit to evaluate the natural trend of GMC, based on a biological approach that accounting height and body mass, as two independent data.

Our results could contribute to ameliorate and finalize the GMC work of PE teachers and sports trainers during the growth of their children. In fact, the knowledge of the crucial period to develop GMC (aged 8 to 10 years old) and the relationship between GMC development with the body mass and PA level could help PE teachers and trainers to better evaluate their children capacities and, consequently, propose activities and training aimed to the correct development of children and their performances through different age and body shape.

## CONCLUSIONS

In conclusion, our finding improved the features of previous literature that suggested an increase in fat body mass is negatively correlated with a rise in GMC level, indicating a positive association between lean body mass with GMC. Lean body mass could be evaluated as a good predictor of GMC in children, especially for teachers, trainers, or sport scientists in talent identification. Furthermore, the period between 8–10 years old is crucial to improve the GMC significantly and subsequently tend towards stabilization from 10 to 13 years old.

### Funding
The authors received no funding for this work.

### Competing Interests
The authors declare that they have no competing interests.

### Author Contributions
- Matteo Giuriato conceived and designed the experiments, performed the experiments, analyzed the data, prepared figures and/or tables, and approved the final draft.

- Nicola Lovecchio conceived and designed the experiments, performed the experiments, authored or reviewed drafts of the article, and approved the final draft.
- Vittoria Carnevale Pellino performed the experiments, analyzed the data, prepared figures and/or tables, authored or reviewed drafts of the article, and approved the final draft.
- Jan Mieszkowski analyzed the data, authored or reviewed drafts of the article, and approved the final draft.
- Adam Kawczyński performed the experiments, analyzed the data, authored or reviewed drafts of the article, and approved the final draft.
- Alan Nevill conceived and designed the experiments, analyzed the data, prepared figures and/or tables, authored or reviewed drafts of the article, and approved the final draft.
- Valentina Biino conceived and designed the experiments, performed the experiments, authored or reviewed drafts of the article, and approved the final draft.

### Human Ethics

The following information was supplied relating to ethical approvals (*i.e.*, approving body and any reference numbers):

This study was approved by the Institutional Ethical Boards of Schools (Prot. 1523; Cod 123.1/6).

### Data Availability

The raw data is available in the Supplemental File.

### Supplemental Information

Supplemental information for this article can be found online at http://dx.doi.org/10.7717/peerj.13483#supplemental-information.

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
