# Peer review of "Gross motor coordination and their relationship with body mass and physical activity level during growth in Children aged 8–11 years old: a longitudinal and allometric approach"

_PeerJ, doi:10.7717/peerj.13483_

## Round 0.1 · original submission · Major Revisions

The article has merit, but some changes must be performed to increase the overall quality.

·

Basic reporting

.

Experimental design

.

Validity of the findings

.

Additional comments

In the introduction, the importance of the study in terms of literature should be explained.
In the introduction, the hypotheses of the study should be emphasized.
In method section, Which sampling method was used in the creation of the sample group in the study. Please explain this.
“The age of the children from the birth date was calculated and subsequently rounded down values” please base it on scientific reference.
“Physical Activity Questionnaire”; Has the reliability of the scale been calculated for this study? It must be calculated. Detailed information should be given about the sub-dimensions and scoring of the scale.
Information about the research design should be given. It should be based on a scientific reference.
Detailed information on data collection should be given. When, where and by whom was the data collected? The reason for choosing the morning hours for the collection of measurements should be stated.
An explanation should be given about the time periods in which the measurements are made each year (October 2017 attended third, fourth, fifth, and sixth grade???). If necessary, the times of the measurements should be schematized in the study.
The sub-dimensions of the KTK battery should be explained in detail.
The discussion section is sufficient, but the use of more recent studies is important for the quality of the article. At the same time, the practical application section should be added.

Reviewer 2 ·

Basic reporting

The article presents clear professional English and the contents are unequivocally exposed. However, it requires some one-off fixes. For example:
lines 59 to 62 - "Barnett et al. (2016) suggesting that variables such as age, sex and weight status affect GMC while Queiroz et al. (2020) found a positive modulation and improvements (in addition to the Body mass index (BMI) in young belonging to a family with a high socio-economic-status family that allows greater possibilities to access in environmental contexts, sports facilities or sports clubs." Review the sentence, as there seems to be a word missing, as the idea is not understood.
line 67 - Do the authors want to refer to sex change?
line 70 - "timing and tempo structured". Didn't they mean structured time?
line 73 - "that, ipso facto,...", please review in English.
line 194 - "IBMI" is the first time these abbreviations appear, please write in full.
References are current and relevant to the topics presented. The structure of the article complies with the journal's instructions, as well as the structure of tables and figures. Data supporting the study has been shared.

Experimental design

The Study is relevant, original, and adequate to the objectives of the journal.
However, the objective of the research, as well as the problem, require further refinement and I even suggest reformulation. It is important to define the problem, in the sense of exposing what literature has not yet demonstrated. Subsequently, build the objective, to respond to this gap. The aim of the study was: "the aim of this study was focused on a longitudinal investigation (three consecutive years) of the GMC levels within the relationships between height and body mass using the allometric scaling analyzes in samples of children aged from 8 to 13 years ." To support this objective, we suggest the inclusion and/or develop some topics in the introduction: importance of body mass for the development of gross coordination; characterization of growth and motor development of these age groups. Additionally, it is suggested that physical activity be included in the objective, as it was a result that was evidenced.
Regarding the title, it was "Gross motor coordination during growth in children: a longitudinal and allometric approach". I suggest that it be reformulated and that it represents the main theme of the study, the relationship between body mass and physical activity in gross motor coordination in a certain period of time…

The investigation was rigorous, not least because the statistical method is considered one of the strengths of the article. Ethical procedures were applied as required.

The methods have been described in detail; however some explanations and suggestions are suggested:
- In the methodology, the dates range from 2017 to 2019. But the results go up to 2020 and do not present data from 2019! A brief explanation of this difference is suggested.
- If boys started the research, for example in the 6th year, after 3 years they are 15 years old… from the point of view of growth and maturation it becomes complicated, as they are already in the puberty phase. Please clarify this aspect.
- There is no reason for choosing this sample: children between 8 and 13 years old. And why not, in children between 3 and 7, who are at a fundamental stage in terms of motor development? Please, substantiate this question.
- In the characterization methodology, the BMI, which is a very subjective measure to be one of the main variables of the study and that characterizes growth, according to the title. However, in the results, in the discussion and in the conclusion, they always refer to body mass, namely fat mass and lean mass. I would like to point out that these measurements were not evaluated, but only height and body mass. Therefore, this topic has to characterize only the variables that were taken into account in the statistical analysis. Please review this part and reformulate the results and discussion accordingly.

Validity of the findings

The impact and novelty of the study were mentioned, however, they require some reformulation, according to what was exposed in the previous points.
All underlying data has been provided; they are robust, statistically sound, and controlled. However, the results indicate that this model defines the factors that best define body size/shape... but they did not measure these variables, such as perimeters, folds... for example to calculate morphology, as was done in the study by Dos Santos, 2018… keep this limitation in mind and do not make the result subjective
Regarding the discussion, it should be reformulated, considering the new reformulation of the title and objective of this study. It is also suggested that they only base the results according to the variables evaluated and not talk about fat mass and lean mass, as it was not applied. Review the entire topic, please.
Still in the discussion, the following explanation/suggestions are suggested:
- about the KTK instrument: could the data not be influenced by the “ceiling effect” of the KTK test? Something that has been demonstrated in the literature. I suggest introducing this topic into the discussion.
- line 237 to 239: This part does not have a link to the text. They are substantiating the children's PA has a positive influence with the GMC, but I suggest that you present the main result before discussing. In addition, Faigenbaum, 2020 is linking physical exercise, specifically a strength training program, and so far only physical activity has been talked about and related.
- lines 250 to 255: They are discussing and substantiating the KTK-3, but it is not the purpose of the study.
- lines 256 to 260: The influence of PA is little explored in the discussion. And it is an important result to be explored and discussed.
- lines 267 to 268: They suggest these ages, because they were the only ones that entered the study… But the previous ages? especially the ages included in the fundamental phase of the movement, according to Gallahue, 2013? Where do children learn FMS and work on motor competence? It would be more pertinent to mention that in this study it was at the lowest ages where motor coordination work is considered and not just between 8 and 10. They are suggesting a window of opportunity, but they did not study the previous ages!

Conclusions should be reformulated in accordance with the changes suggested above. Mainly the question of suggesting a window of opportunity to stimulate gross motor skills.

Additional comments

General comments
The purpose of this study was focused on a longitudinal investigation (three consecutive years) of the GMC levels within the relationships between height and body mass using the allometric scaling analyses in samples of children aged from 8 to 13 years. It is a very important study, adds important evidences for the literature and well-structured, so authors should be commended. However it is suggested to go deeper into the introduction, methodology and discussion, and to reformulate the title, objective and conclusion.

Reviewer 3 ·

Basic reporting

Well structured article well structured.
Use of current references.

Experimental design

No comment.

Validity of the findings

No comment

Additional comments

The introduction should be restructured, especially the penultimate paragraph that presents information that should not be included in the introduction.
Procedures can be more detailed.
They could mention the limitations of the study.

---

## Round 0.2 · accepted · Accept

The authors performed the necessary changes to improve the article. The reviewers agree on the acceptance.

·

Basic reporting

To me, I examined the revised file and rebuttal letter carefully.. Congratulations to the authours. They made big changes in the article.. Also, they made all corrections which I gave.. I think that the article is ready for publication..

Experimental design

To me, I examined the revised file and rebuttal letter carefully.. Congratulations to the authours. They made big changes in the article.. My revisions were related to improving the methods section.. Also, they made all corrections which I gave.. I think that the article is ready for publication..

Validity of the findings

To me, I examined the revised file and rebuttal letter carefully.. Congratulations to the authours. They made big changes in the article.. Also, they made all corrections which I gave.. I think that the article is ready for publication..

Additional comments

To me, I examined the revised file and rebuttal letter carefully.. Congratulations to the authours. They made big changes in the article.. Also, they made all corrections which I gave.. I think that the article is ready for publication..

Reviewer 2 ·

Basic reporting

no comment

Experimental design

no comment

Validity of the findings

no comment

Additional comments

The purpose of this study was aimed to explore the developmental changes in GMC during childhood, having controlled for the differences in children's body size and shape using a longitudinal, allometric scaling methodology. It is a very interesting study, adds important evidence well-structured and works well for its clearness, so authors should be commended. The main problems were resolved with manuscript reformulation, and I agree with all changes made to the article.